# Probing the Inductive Bias of Neural Networks through Learning Random Cellular Automata

## Abstract

In this paper, we empirically examine whether the inductive bias of deep networks can be linked to structural properties of dynamical systems inspired by physics, such as symmetry, locality, and coarse-grained observation of outcomes. To explore this question, we generate "toy universes" by sampling random cellular-automaton rules that satisfy these constraints, and train convolutional neural networks (CNNs) to predict their evolution under three experimental factors: temporal coarse-graining, spatial pooling, and a structured (low-entropy) initial state. Throughout, we measure each network's average generalization performance relative to a baseline. While classical constraints such as symmetry and locality are necessary, they alone are not sufficient for learnability. However, when we account for the perturbation sensitivity of the target function, we observe a strong negative correlation with learnability. Further, using a structured (low-entropy) initial state leads networks to favor coarser macroscopic patterns over details.

## 1 Introduction

In recent years, deep learning has been the driving force that has for the first time allowed applications of machine learning to complex, "real-world" problems. From a conceptual perspective, not only is the remarkable generalization performance on complex tasks intriguing; the fact that the same class of techniques worked on a large variety of problems and data, from images and music to language and quantum chemistry with only minor and generic domain-specific adaptations [17], might be equally surprising. As statistical learning requires a strong inductive bias to generalize [44, 10, 16], with a gap exponential in input dimensionality [26], this implies that all these different kinds of data must share a common and, mathematically speaking, highly specific statistical structure. From this perspective, understanding the *prior of deep learning* relates to finding common structure in (most) naturally forming patterns. Once the existence of a common inductive bias in deep learning became apparent, identifying the principles behind it – in other words, characterizing the "prior of deep learning" – rose to a major scientific question. And it suggests a compelling hypothesis: Do the laws of physics already bias pattern formation in a way that enables intelligence as such, and deep learning in particular? More specifically, might some of the *structural principles* that physics itself follows [26] be responsible for generalizable pattern recognition? These principles include e.g. spatio-temporal *symmetry*, *locality* or entropy increasing with time (e.g. as reversible evolution starting in low-entropy initial conditions). Unsurprisingly, finding hard evidence for such a link has proven to be non-trivial. In this paper, we try to approach the question of linking these structural principles to learnability by deep neural networks through an experimental approach: We fix some of the structural principles and then build a randomly chosen "toy universe" that follows those and try to learn to predict patterns in the resulting system via deep learning. This begs the question of which superset of "universes" to and how to define "learning the patterns". At this point, we have to make rather strong simplifying assumptions:

**Laws & Universes:** In order to make a study tractable, we have to restrict ourselves to a discrete set of candidate laws, and we have chosen cellular automata (CAs) with binary cells as this could be considered a minimal model system. Obviously, this serves as an analogon for a qualitative understanding, not as a model to describe actual real-world physics.

**Learning Patterns:** Similarly aiming at a simple formalization of "learning" pattern formation, we pick the task of simply predicting the future evolution of given initial conditions. We assume perfect knowledge of the initial state (input) but study various degrees of temporal and spatial coarse-graining (on the output side) to study scale effects.

We also have to be careful in evaluation and interpretation of the results:

**Performance:** For evaluation, we measure generalization performance, not memorization. To quantify success, we relate performance to a simple baseline to exclude being misled by a trivial success of rules that do not generate meaningful patterns, the increased frequency of which can be a side effect of constraining the dynamics. We also make sure to train in a regime that would permit identifying the (purely randomly chosen) rules when the additional coarse-graining structure was fully known and modeled by the learning system (which it is not), as it is obviously not possible to guess random bits of information with any conceivable piece of prior information.

**Level of Emergence:** Our experiments use only moderate numbers of timesteps and moderate coarse-graining, i.e., they involve only a small number of computational steps. Thus, they could be interpreted as an analogon of a rather simple, macroscopic physical system with a small number of interactions and interacting parts. Describing structure formation along large scale differences and time scales would bring up insurmountable problems of undecidability and lack of statistical power.

Despite requiring fundamental computational compromises, our study reveals some interesting insights: First, we can easily convince ourselves that *locality and symmetry are necessary* in the setting outlined above, but, empirically, turn out *not sufficient*. Including *sensitivity* of the overall computation to perturbations of initial conditions as a parameter shows a significant correlation with training success, declining with sensitivity and timesteps, but still not being sufficient. Further, we investigate the impact of structured "low-entropy" initial conditions: Here the network is able to predict the coarse-scale "shape" of the output - corresponding to the spread of information in the automaton, but is often unable to predict the fine-scale " texture", suggesting that neural networks are able to fit low information subsets of a problem when possible, while ignoring more complex parts.

Overall, by using a novel experimental approach of building "random toy universes" under constraints on its "physics", we can show that the combination of principles examined is strongly correlated, but not fully sufficient for learnability. The persistent simplicity bias could point towards a hidden, yet unknown principle of favoring stability in structure formation.

## 2   Related Work

**Universal Priors:** The paradox of universal induction has intrigued researchers for a long time, with the impossibility result of "no-free-lunch" [44, 16] generally resolved by an appeal to a variant of Occam's razor [37, 38, 35, 27, 19], which demands concise coding and breaks the symmetry by assuming a mathematical language. This does not relate to natural dynamics in an obvious way [42].

**Priors of Deep Learning:** Prior knowledge can be incorporated explicitly, e.g., by exploiting symmetry [14, 8] or multi-scale modeling (e.g., via pooling [46]). But even for a basic fully connected MLP, biases are known. Prominently, they learn low-frequency features more quickly [34]. Using a tangent-linear model (NTK) of network training [31, 20], this bias can be understood as decaying eigenvalues of kernel-eigenfunctions [7]. This tangent-spectral picture also explains phenomena such as double descent [4, 3], grokking [24] and, to some extent, adversarial examples [40]. The notion of *sensitivity* (susceptibility of a function to small input perturbations [21]) is closely related to the spectral bias [41]: A correlation between discrete sensitivity and generalization in neural networks has been established early on [12], and similar findings hold in a large empirical study for various notions of sensitivity (including continuous ones, such as sharp minima) [32]. The findings have been replicated for recurrent architectures and transformers, which showed an even stronger bias [6]. Again, an NTK model provides an explanation [41] with an explicit link between continuous low-frequency bias and discrete sensitivity, as we use in our paper. Given what we already know

about sensitivity, the main insight in our paper is that sensitivity seems to be orthogonal to the other structural principles of physics and highly correlated with, but not sufficient for generalization.

In a different line of work, Mingard et al. [28] propose that the prior of deep networks can be characterized as simple Bayesian sampling from the initialization distribution in function space, with good empirical matches in a Gaussian approximation. They also show that this induces a low-complexity prior [29]. While constituting a big step towards a better understanding of the inductive bias of deep learning, it does not address the question of a connection to dynamical processes.

**Links to Physical Dynamics:** Machine learning as a field has been influenced strongly by concepts from natural science and physics [36, 47]. In terms of causal links far-reaching hypotheses have been proposed, such as linking predicting dynamics to self-preserving intelligent structures [13], or dualities of learning and fundamental physics [2]; however, it remains challenging to prove or disprove models at this scope. The influential article by Lin et al. [26] enumerates concrete links between physical models and structural properties of deep networks at a formal level, in particular showing their ability to encode low-order polynomial Hamiltonians, exploiting symmetry for compactness, and re-addressing renormalization (also addressed elsewhere, e.g. [9]) for bridging scales, but still leaves open how complex emergent structures are captured.

**Cellular automata (CAs)** have long been used as model systems for physical dynamics, both in a concrete sense of discretization of fundamental physics [18] as well as merely an abstract analogon, as in our paper. Connections between sensitivity and complexity measures such as entropy and Lyapunov exponents have been studied by Langton [25], referring to Wolfram's foundational categorization [43]. CAs have also already been used as model system in studying neural networks: Wulff and Hertz [45] learn single timesteps and already identify that chaotic CA rules posed significant learning challenges. Gilpin [15] demonstrate theoretically and empirically that CNN architectures are capable of explicitly encoding the local rules underlying CA dynamics, asserting that given sufficient training data, CNNs can precisely replicate CA update rules. Springer and Kenyon [39] show that the Turing-complete Game-of-Life is hard to learn in a setting of temporal coarse-graining. Elser [11] follow-up by design a training protocol to sample good training examples. Aach et al. [1] explored generalization across multiple CA rules, finding that CNNs could partially generalize to unseen configurations and even unseen rules within certain constraints. On the flip-side, Neural Cellular Automata [30] demonstrate that strictly local iterative rule application can be learned that results in highly complex patterns. Bhamidipaty et al. [5] use model systems for algorithm evaluation, including CAs, but their work does not aim at links to physics. Our study differs to previous work in its approach to study the connection between physically-motivated constraints and learnability.

# 3 Methods

## 3.1 Cellular Automata as Discrete Dynamical Systems

We model discrete dynamical systems using Cellular Automata (CA). A CA describes the evolution of a state defined over a discrete grid and time. Formally, let the state be a function $s : \Omega \times \mathbb{Z} \to \mathbb{B}$, where $\Omega \subset \mathbb{Z}^d$ is the spatial grid (here $d = 2$), $t \in \mathbb{Z}$ is discrete time, and $\mathbb{B} := \{0, 1\}$ represents the binary state of each cell. The evolution is governed by a local transition function $f : \mathbb{B}^k \to \mathbb{B}$ applied synchronously to all cells:

$$s(r, t + 1) = f(\mathcal{N}(r, t)) \tag{1}$$

where the neighborhood $\mathcal{N}$ contains $k$ cells spatially adjacent to $r$ and $r$ itself at time $t$. The function $f$ and an initial condition $s(\,\cdot\,, t_0)$ determine the system's entire trajectory. For convenience, we also use $f$ to denote the map $s(\,\cdot\,, t) \to s(\,\cdot\,, t + 1)$, obtained by applying $f$ to every cell at the same time. Specifically, the state $s$ is defined on an $H \times W$ grid with binary values ($\mathbb{B} = \{0, 1\}$) and torus topology (periodic boundary conditions). We use a $k = 9$ Moore neighborhood (the $3 \times 3$ square centered on the cell $r$). The transition function $f : \mathbb{B}^9 \to \mathbb{B}$ maps each of the $2^9 = 512$ possible neighborhood states to a next state for the central cell. We mostly use outer-totalistic CAs, where only the sum of the neighborhood and the value of the central cell are used to calculate the next state. We do so as outer-totalistic CA have a higher chance of exhibiting interesting behavior and have a lower complexity. Nonetheless, control experiments with fully general rules did not have qualitatively different outcomes and results of these experiments are provided in the Appendix. Rules are sampled uniformly at random from the entire rule-space ($\#\mathcal{F} = 2^{18}$ for outer-totalistic automata) for each experiment, unless otherwise specified.

## 3.2 Coarse-Graining

To model realistic scenarios where observations of dynamical systems are typically imperfectly resolved in time and space, we introduce two coarse-graining procedures: temporal and spatial. These allow us to study how varying levels of granularity influence learnability by deep neural networks.

**Temporal coarse-graining** is implemented by repeatedly composing the CA update function $f$ : $\mathbb{B}^{H \times W} \to \mathbb{B}^{H \times W}$ over multiple discrete timesteps. For a given temporal scale $T$, we define the temporally coarse-grained mapping $f^{(T)}$ as:

$$f^{(T)}(x) = f(f(\dots f(x))) \quad (T \text{ times}). \tag{2}$$

By training neural networks directly to predict $f^{(T)}(x)$ from initial conditions $x$, we test the networks' ability to handle compounded dynamics and identify whether there are temporal thresholds beyond which the complexity becomes unlearnable.

**Spatial coarse-graining** reduces the spatial resolution of the CA state after evolution. In our experiments, we implement spatial coarse-graining through a majority-voting pooling operator $P_S$, which aggregates each overlapping $S \times S$ neighborhood of cells into a single binary value. Formally, spatial coarse-graining transforms the evolved CA state as follows:

$$x \mapsto P_S(f^{(T)}(x)). \tag{3}$$

Spatial coarse graining accounts for observations at limited resolution. By varying the spatial pooling factor $S$, we can study how spatial resolution influences neural network learnability (positively or negatively). Together, these two coarse-graining procedures allow us to characterize how neural network learnability depends on the level of detail available in both temporal and spatial domains.

## 3.3 Fundamental Constraints: Locality and Symmetry

Our choice of CAs implicitly incorporates structural constraints analogous to those in physical systems, namely locality, symmetry, and complexity. We discuss each below:

**Locality:** The update rule $f$ depends only on a small, spatially contiguous neighborhood ($k = 9$). This restriction is crucial from an information-theoretic perspective, as in the fully general case, the number of possible transition functions grows doubly-exponentially by $2^{2^k}$ with the neighborhood size $k$ [43]. Less locality would thus require an exponentially larger amount of training data to identify. Our CNN architecture (Sect. 3.5) incorporates this prior explicitly.

**Symmetry (Spatial and Temporal Invariance):** The same transition rule $f$ is applied identically at all spatial locations $r$ and all timesteps $t$. This spatio-temporal symmetry is essential for generalization in our setting of CAs. Generally speaking, invariance forms the foundation of inductive reasoning as it permits reproducible experiments (or identically distributed training data, in statistical terms). Our CNN models spatial symmetry over the full receptive field (growing with temporal coarse-graining) but does not impose temporal symmetry within the network (i.e., weights are not shared across layers). The 90° rotational/reflective symmetry of outer-totalistic CAs is not exploitet.

**Complexity:** Empirical studies of neural networks demonstrate a strong inductive preference towards simpler functions, often described as a simplicity bias related to Kolmogorov complexity $K(f)$ [29]. Within our experimental framework, the complexity of the learned functions — specifically, the mapping from initial states to evolved states — is naturally constrained by the simplicity of the underlying CA transition rule. As a result, the functions we attempt to learn have significantly lower complexity than arbitrary binary functions defined on grids of comparable size. This inherent simplicity ensures that observed differences in learnability predominantly reflect meaningful structural properties rather than information-theoretic limitations.

## 3.4 Training Data and Initial Conditions

Training data consists of pairs $(s_0, s_T)$ where $s_0 \in \mathbb{B}^{N \times N}$ is an initial state and $s_T = f^{(T)}(x)$ is the state after $T$ timesteps under the chosen rule $f$. We generate initial conditions $s_0$ using three distinct procedures to probe different aspects of network learnability:

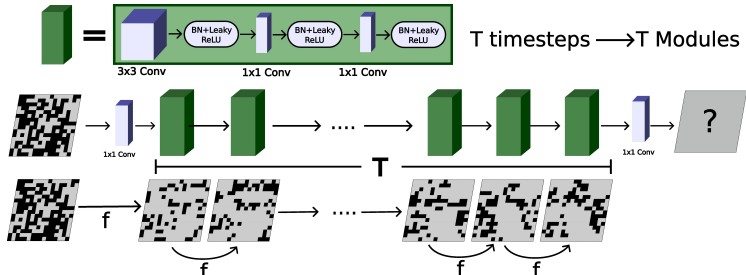

Figure 1: Schematic overview: The CNN architecture used consists of T blocks, the same amount as timesteps to be modeled. This guarantees that it is able to fit a given function at the given timescale. We use LeakyReLU and BatchNorm after each convolutional layer except for the last one.

- **Random initialization:** Each cell $s_{0_{ij}}$ is sampled independently and uniformly from $\mathbb{B}$, i.e., $P(s_{0_{ij}} = 0) = P(s_{0_{ij}} = 1) = 0.5$. This procedure serves as a standard baseline to measure generalization under uniform randomness.

- **Naturalized states:** To produce a "steady-state" configurations, we first initialize states randomly as in (1), and then evolve them by applying $f$ for a small number of steps $t$. The resulting training data consists of pairs $(s_t, s_{T+t})$. This approach assesses the network's capacity to predict CA dynamics from states closer to their natural, evolved distributions.

- **Localized initialization:** Initial states are generated randomly as in (1), but with a fixed border of width $T$ pixels on each side set to $0$. This construction explicitly tests the influence of spatial gradients in information density on learnability.

The first two procedures correspond to constraints of maximum entropy, with the second one modeling thermal equilibrium. The third represents constraints to the initial state of lower entropy.

## 3.5 Network Architecture and Training

Our model architecture is a fully convolutional network designed to respect the locality and spatial invariance of the CA dynamics. To predict $T$ steps ahead, the network consists of $T$ sequential blocks, giving the opportunity to model one timestep per block. Crucially, these blocks do *not* share weights, allowing the network to learn potentially distinct intermediate representations at each step.

Each block consists of:

1. A $3 \times 3$ Conv2D layer, modeling interactions within the Moore neighborhood. Circular padding is used to maintain spatial dimensions and periodic boundary conditions.

2. Followed by two $1 \times 1$ Conv2D layers, increasing representational expressivity while preserving locality.

3. BatchNorm and LeakyReLU activations between each convolutional layer for stabilization and non-linearity.

The network's final prediction is obtained through a $1 \times 1$ convolutional layer outputting two channels, corresponding to binary logits for each cell. All intermediate convolutional layers employ $128$ feature channels. A schematic overview is provided in Figure 1. Note that structuring the network as one block per timestep guarantees that the network is able to model the function we are trying to learn, as a single block, in principle, has the ability to model a single timestep. Networks are trained using the Adam optimizer [23] with a learning rate of $4 \times 10^{-4}$ and a binary cross-entropy loss. Training is performed for $4096$ iterations with a batch size of $64$, using patches of size $16 \times 16$ or $32 \times 32$ cells depending on the specific experiment. This procedure makes $T = 1$ treatments (i.e., the rules as such) easily learnable, but is non-trivial for larger $T$ (experimentally visible in Fig.2 for T=2,3,...).

Hyperparameters such as learning rate, batch size, and channel depth were selected based on smaller-scale preliminary experiments to ensure stable and effective training.

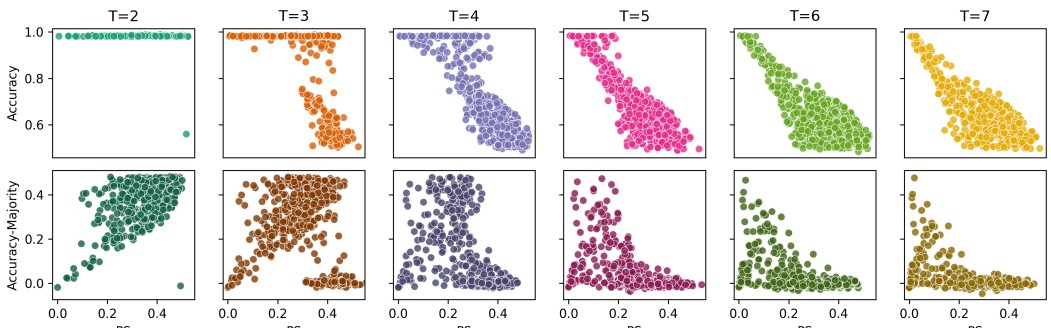

Figure 2: Results of different training runs for different timescales. From left to right, coarse-graining increases from $T = 2$ to $T = 7$. First row shows accuracy of the trained network, second run shows accuracy difference compared to a simple majority classifier. We can see that with increased temporal coarse-graining it becomes more difficult to learn functions with higher perturbation sensitivity.

## 3.6  Evaluation Metrics

To quantify learnability, we monitor performance metrics of the neural networks:

- **Pixel-wise Accuracy:** The fraction of correctly predicted cell states.
- **Accuracy Gain over Baselines:** We compare network accuracy to a baseline of predicting the majority class. We chose this baseline as networks frequently defaulted to constant predictions when training failed, allowing for an easy way to compare to the "failure case". In one scenario, we additionally include comparisons to a logistic regression classifier.

We also measure the **Perturbation Sensitivity (PS)** of $f^{(T)}$ as a computationally feasible proxy for the complexity of the mapping. PS quantifies how sensitive the system evolution is to small input changes by measuring the average effect of flipping a single bit in the input state, formally:

$$\text{PS}(f^{(T)}) \approx \frac{1}{N} \sum_{n=1}^{N} \mathbb{E}_{i \sim \text{Unif}(1, HW)} \left[ \frac{1}{HW} \sum_{j=1}^{HW} |f^{(T)}(x_n)_j - f^{(T)}(x_n^{(i)})_j| \right] \tag{4}$$

where $x_n^{(i)}$ denotes the state $x_n$ with its $i$-th bit flipped. Unlike alternative complexity metrics (e.g., Kolmogorov complexity, Lempel-Ziv complexity), PS is efficiently computable and directly measures the sensitivity relevant to prediction robustness [12].

We conduct a broad experimental sweep across various rules $f$, temporal depths $T$ and spatial coarse-graining sizes $S$, examining relationships between PS and predictive accuracy for standard and naturalized initialization. We also systematically compare performance under globally randomized versus localized initializations.

## 4  Experiments

### 4.1  Learnability and Temporal coarse-graining

To investigate the effect of temporal coarse-graining on learnability, we sample $512$ random outer-totalistic CA rules $f$. For each rule, we train separate CNNs to predict the state after $T$ timesteps, where $T \in \{2, 3, 4, 5, 6, 7\}$ for a total of $3072$ training runs, taking $\sim 15$ hours on an NVIDIA GeForce RTX 4090. We measure validation accuracy achieved by the network after $4096$ training iterations and compute the Perturbation Sensitivity (PS) of the corresponding $T$-step function $f^{(T)}$.

Figure 2 shows that predicting short time horizons ($T = 2, 3$) is feasible for most rules, with networks typically achieving high accuracy, significantly outperforming the majority-vote baseline, and often reaching near-perfect ($\sim 100\%$) prediction accuracy. However, as the prediction horizon $T$ increases, learnability deteriorates rapidly. For $T \geq 4$, highly sensitive rules become effectively unpredictable by the CNN, with accuracy collapsing towards baseline performance. Moreover, Figure 2 illustrates

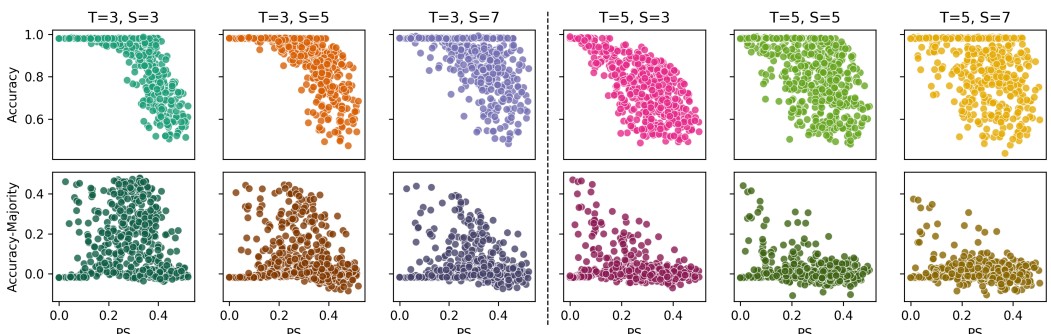

Figure 3: Results of different training runs for different spatial coarse-grainings. From left to right, coarse-graining increases from $S = 3$ to $S = 7$ with temporal coarse-graining of $T = 3$ left and $T = 5$ right. First row shows accuracy of the trained network, second run shows accuracy difference compared to a majority classifier baseline. While accuracy seems to increase with larger coarse-graining, comparison with the baseline shows that this is mainly driven by rule simplification.

a clear relationship between PS and the accuracy achieved by the CNN. Accuracy consistently decreases as PS increases, especially at higher temporal scales ($T$). This trend remains even when measuring accuracy relative to the baseline. Networks consistently outperforming the baseline appear predominantly at lower PS values, with the maximum PS at which CNNs outperform the baseline decreasing as the temporal scale increases. At large $T$, high sensitivity excludes outperforming the base-line. Low sensitivity is not sufficient but increases the chance of learnability.

We repeated these experiments using naturalized initial conditions, resulting in an additional 3072 training runs, taking an additional $\sim$ 22 hours. Results were qualitatively similar, confirming the robustness of the observed trends and the impact of perturbation sensitivity on predictability.

## 4.2 Learnability and Spatial coarse-graining

To evaluate how spatial coarse-graining impacts learnability, we again sample 512 random outer-totalistic CA rules. For each rule, we train networks to predict CA evolution under combinations of temporal $T = 3, 4, 5$ and spatial coarse-graining scales $S = 3, 5, 7$, resulting in a total of 4608 training runs, taking $\sim$ 33 hours of training time. Because spatial coarse-graining effectively enlarges each cell's receptive field, we increase CNN depth by adding $1, 2$, or $3$ convolutional blocks for spatial scales $S = 3, 5, 7$, respectively. Results from spatial coarse-graining experiments differ notably from those obtained with temporal coarse-graining alone, as can be seen in Figure 3. While absolute accuracy generally increases with greater spatial coarse-graining, improvements relative to the majority-vote baseline are much less pronounced. Indeed, as spatial coarse-graining increases, the baseline predictor becomes inherently stronger, as it benefits from pooling, leaving less possibilities for CNN improvement. As a result, CNNs rarely outperform the baseline for functions exhibiting high perturbation sensitivity. Still, the relationship between lower PS and improved relative performance remains apparent, but weaker than in purely temporal experiments.

Repeating the spatial coarse-graining experiments with naturalized initial conditions (another 4608 training runs, $\sim 50h$ compute time) yields largely similar results, with one notable exception: a distinct cluster of rules emerges exhibiting low PS and high accuracy relative to the baseline. Analysis indicates that this cluster consists of rules converging to fixed points, thereby simplifying the prediction task to an identity mapping. Replacing the majority-vote baseline with logistic regression removes this cluster, producing results similar to those obtained with randomly initialized states.

## 4.3 Learnability and Localized Initial Conditions

To examine how localized entropy influences learnability, we perform experiments comparing training outcomes under localized initial conditions (as defined in Section 3.4). As in previous sections, we sample 512 random CA rules and train CNNs to predict their states after temporal coarse-graining with horizons $T = 4, 5, 6$ for a total of 1536 runs. However, instead of initializing the entire patch uniformly at random, we use a structured initialization, fixing a border of width $T$ pixels on each side

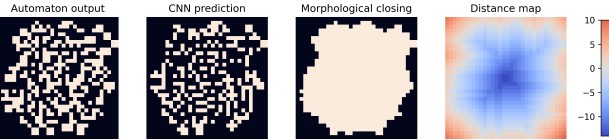

Figure 4: How distance to objects are calculated. To the left we see the automaton output and the CNN prediction. We apply a morphological close to the automaton output and calculate signed distances to the resulting object. Positive distances correspond to the exterior (red), negative ones to the interior (blue).

to zero. Because such a setup yields very small central regions on $16 \times 16$ patches, we perform the experiments on larger $32 \times 32$ patches, taking $\sim$20 hours of compute time.

While the overall prediction accuracy does not clearly improve under these localized initializations, examining predictions visually reveals an interesting structural phenomenon. The network is often able to predict the overall "shape" of the evolved pattern correctly, achieving significantly higher accuracy along the edges of these shapes compared to their interiors. While improved accuracy on the outermost edges of the $32 \times 32$ patch might be trivially expected, the phenomenon is present even for smaller shapes formed by localized initializations.

To quantitatively evaluate this observation, we analyze prediction accuracy relative to spatial distance from the shape boundary. Specifically, given a prediction and a correct label, we first run a morphological closing operation on the label. We then calculate the signed $l_1$ distance from each cell to this shape, assigning positive distances to cells outside the shape and negative distances to cells inside (see Figure 4 for an illustration). For each prediction-label pair, we calculate average accuracy for each distance, and aggregate results across 32 independent prediction-label instances per rule.

Examples (see example visualizations in Figure 6, and aggregated results over 100 randomly selected rules in Figure 5) confirm the observed trend. CNN predictions consistently approach near-perfect accuracy ($\sim 100\%$) at positive distances (outside the predicted shape) but show a rapid and consistent decline at negative distances (inside the shape). A large fraction of prediction accuracy thus stems from the CNN's ability to infer how far spatial information can propagate from the local initialization, though the exact pattern details remain difficult for the network to predict accurately. Consequently, the CNN reliably matches the overall output shape—and occasionally simpler interior details, but struggles with complex fine-grained internal structure.

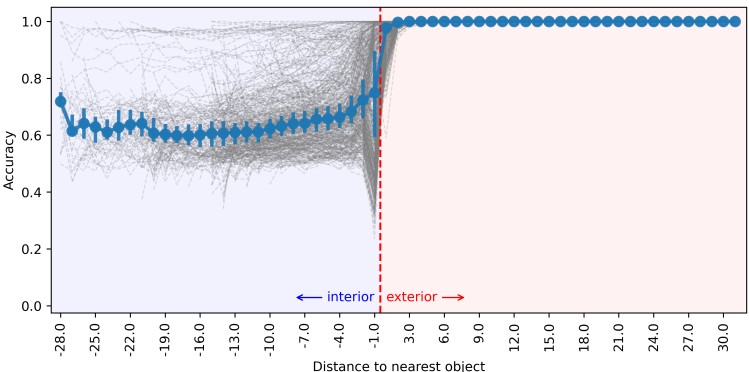

Figure 5: Accuracy by distance for 400 randomly drawn rules. Each gray line corresponds to the mean accuracy for the given distance in one such example, calculated over a batch of 32 samples. Blue dots correspond to the median accuracy over all rules, with blue bars representing a 25% quartile around the median. The light-blue region corresponds to the interior, the light-red to the exterior. We can see that on average a CNN loses substantial amount of accuracy on the edge of the automaton, and then continues to lose accuracy further in the interior. In the innermost parts, the models regain accuracy, but this is partially an artifact of few examples with large negative distances.

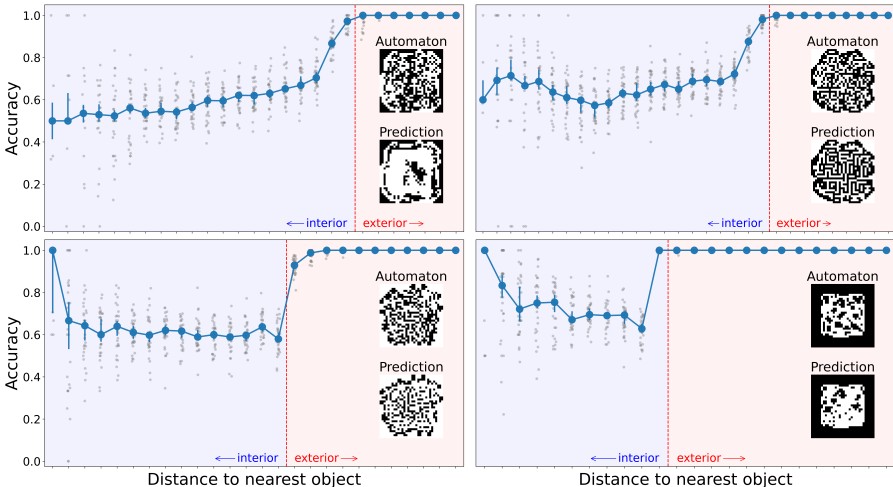

Figure 6: Accuracy by distance to closure over a batch of 32 examples for 4 different automata. Each blue dot corresponds to the median accuracy in one such example, while the blue bars represent the surrounding 25% quartile. We also show a single output of the automaton and the corresponding network prediction. While the network is able to predict the outer edges of the automaton, it is unable to infer the interior with the same accuracy. More examples are provided in the Appendix.

## 5  Discussion and Conclusions

Our experiments using CAs as model systems provide several insights into the conditions influencing DNN learnability of dynamical systems. First, fundamental physical constraints like locality and spatio-temporal symmetry, while necessary and naturally embodied by both CAs and CNNs, are insufficient to guarantee predictability. CNNs often failed to learn the evolution of even simple, local, and symmetric CA rules beyond short time horizons ($T \approx 3$–4) when trained on high-dimensional, randomly initialized inputs. Second, system sensitivity, measured by PS, correlates with learnability in a scale-dependent way. At longer time horizons, high sensitivity leads to more frequent prediction collapse. However, this relationship is not linear or complete, suggesting that other aspects of system complexity, such as long-range correlations, emergent macrostructures, or proximity to criticality (e.g., Wolfram's classification [43] or Langton's $\lambda$ parameter [25]), may also shape learnability boundaries. Third, the structure of the initial state has a strong impact on learnability. We see clear differences in accuracy along the "edges" of predicted structures under localized initial conditions, which suggests that predictive failure in the globally random case is due to the difficulty of tracking numerous signals across the entire receptive field. This implies that not just the rule's complexity, but also the distribution and organization of the input state, play a role in determining learnability. These findings are consistent with challenges observed in other domains, such as long-term prediction in chaotic systems [33], and the success of methods focusing on localized or sparse interactions.

Limitations of this work include the focus on 2D binary CAs, which are a simplification of continuous physical systems, and the specific CNN architecture used. Future work could explore different CA types (e.g., continuous-valued, higher dimensions), alternative complexity metrics, and architectures like Recurrent Neural Networks (RNNs) or Transformers adapted for spatio-temporal data. Our setting also does not allow the exploration of some further structural principles of classical physics, such as reversability (as testing whether a random CA is reversible is undecidable [22]) or conservation laws (which do not have a straightforward analogon in the discrete case).

In conclusion, this study underscores that DNN generalization for dynamical systems depends on the interplay between the system's rules (locality, symmetry, intrinsic complexity), the structure of the states encountered (localized vs. global activity), and the network's architectural priors. Simply matching basic symmetries like locality is not enough. The effective complexity presented by the data itself is a key determinant of learnability. While sensitivity is a known proxy highly correlated with learnability, it is not sufficient, suggesting that we are still missing a more specific measure.

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
