# OpenReview forum: "Probing the Inductive Bias of Neural Networks through Learning Random Cellular Automata"
_NeurIPS.cc/2025/Conference — Submitted to NeurIPS 2025_

### Official Review · Reviewer_rBzC · 2025-06-30

**Clarity:** 3
**Significance:** 2
**Originality:** 2
**Rating:** 4
**Confidence:** 4

**Summary:**

This paper investigates whether the inductive bias in neural networks is sufficient to ensure the learnability of physical systems that follow corresponding structural principles. The main finding is that inductive bias alone is not sufficient in the 2D cellular automata examples; moreover, the authors observe a negative correlation between learnability and perturbation sensitivity.

**Questions:**

1. *Clarity in Figure 3 (Spatial Coarse-Graining)*. In the spatial coarse-graining experiments (Figure 3), the authors state that accuracy increases with spatial scale. However, this trend is not visually obvious. While it appears that more points cluster near 1, the heavy overlap of data points makes it hard to discern the average behavior. The authors may consider a better representation of the trend.

2. *Coarse-Graining Scales*. As the spatial coarse-graining scale increases, the density of while and black pixels (1s and 0s) can change. At a large scale, one color can domiante which makes the prediction task has a higher null-accuracy. This is analogous to the renormalization group flow in the Ising model, where only at the critical point do we obtain a nontrivial fixed point. The authors could discuss for example how the density of a pixel, the spatial frequency of patterns changes with coarse-grain scales, and whether a possible trivialization affects the interpretation of the results.

3. *Clarification of Section 4.3 Experiments (Interior vs. Exterior Regions)*. In Section 4.3, the exterior region appears as uniformly black pixels, making the prediction task relatively easy in that area. The authors should clarify the motivation behind this experiment. Specifically, whether the higher accuracy in the exterior region is merely a consequence of the low-frequency (or low-complexity) signal present there, or if it reflects a deeper connection to the use of regularized or structured initial conditions.

**Ethical Concerns:**

["NO or VERY MINOR ethics concerns only"]

**Final Justification:**

The learnability problem addressed in this paper is important for scientific machine learning, such as neural network-based modeling of PDEs. In their rebuttal, the authors clarified how their contributions relate to prior work. The question of whether inductive bias alone is sufficient to guarantee learnability is a meaningful and worthwhile direction of study. Moreover, the experimental investigation presented in the paper provides a valuable foundation for future research.

**Limitations:**

Yes

**Quality:**

3

**Strengths And Weaknesses:**

**Strengths**

Whether an appropriate inductive bias in learning models is sufficient to capture the underlying physics in real data is a fundamental question in scientific machine learning. This paper addresses the issue by using cellular automata to design well-controlled experiments for systematic evaluation. The paper is well-written, and the main ideas are clearly presented.

**Weaknesses**

1. *Comparison with Prior Work*. Similar experiments were conducted by Wolfram on 1D cellular automata, where long-term prediction was found to be infeasible due to computational irreducibility. See, for example, his blog post “Can AI Solve Science”. Based on those results, it is intuitive that long-time prediction in 2D systems may also be infeasible. The authors should clarify what new insights or conclusions this work offers beyond previous studies, especially in light of those known limitations.

2. *Interpretation of Results*. Although computational irreducibility is a deep theoretical issue that may resist simple conclusions, the paper would benefit from offering more intuitive interpretations of the results. Even informal explanations could help readers better grasp the implications of the findings.

---

> ### Author Rebuttal · Authors · 2025-07-31
>
> We sincerely thank the reviewer for their thoughtful comments and insightful observations, particularly regarding prior work and the clarity of our experimental interpretation. These points help us strengthen the framing and broader context of our contributions.
>
> #### Comparison with Prior Work and Computational Irreducibility
>
> We appreciate the reviewer highlighting the connections to prior studies on computational irreducibility, notably Wolfram’s work on 1D cellular automata. While computational irreducibility emphasizes the fundamental impossibility of simplifying dynamical predictions in general, our study focuses specifically on the neural network's inductive biases. We investigate scenarios involving relatively short transient processes and limited spatial scales to clarify how constraints such as symmetry, locality, and perturbation sensitivity specifically affect neural network learnability. Our experiments intentionally avoid large $T$ regimes that would potentially approach undecidable dynamics, focusing instead on clearly delineated "small-scale" pattern formation mechanisms to identify necessary and insufficient principles for learnability. We also designed our convolutional architecture to use similar computation steps as the automaton. Each of the T modules used in these architectures is able to fit a single timestep of almost all automata (as can be seen for $T=2$ as an example), implying that the full network with $T$ modules in principle possesses the capability of modeling the function for arbitrary $T$.
>
> #### Interpretation of Results and Experimental Clarifications
>
> We thank the reviewer for pointing out potential ambiguities in the interpretation of our spatial coarse-graining and localized input experiments. Indeed, coarse-graining increases baseline accuracy by simplifying the underlying pattern. However, we did consider this effect in our evaluation already: It is addressed by focusing on accuracy gain relative to the majority-vote (or linear) baseline. This relative measure clarifies how much meaningful information the network learns beyond trivial averages.
>
> We will emphasize this point more clearly in the revision. We also did find the analogy to renormalization of an Ising system quite enlightening and should mention this in our revision, as it provides a intuitive perspective on the findings: In both an “inactive” and “fully chaotic” regime, the network cannot exceed a simple averaging base line; the interesting effects might be located at the transition between the “boring” and the “random regime”, where actual coarser-scale structures form.
>
> Regarding localized initial conditions, we have partially answered this in response to Reviewer uiey. But to be more specific: The CNN is in many cases able to correctly infer the shape of the final state, which is a complex function given by the input initialization. Predicting the exterior as black is trivial and expected, partially due to the low-complexity there.
> But knowing where this exterior begins exactly is a hard task. The results here reflect the earlier results, but in a single function: Subtasks with low PS (the exterior) are easy to learn, while subtasks with high PS (the interior) are hard to learn. The boundary is barely learnable, so we see some success here, but this is followed by a quick and sharp drop in accuracy.
>
> We will better clarify the motivation behind this experiment in our revised manuscript, explicitly connecting it to our broader claims about PS and complexity.
>
> We will also further enhance the clarity of visual presentations, including improving the interpretability of scatter plots as suggested.

---

> > ### Comment · Reviewer_rBzC · 2025-08-01
> >
> > I appreciate the authors’ detailed responses to my questions. The learnability of neural networks for complex dynamical systems is a fundamental topic. Based on the authors’ explanation, I agree that analyzing whether strong inductive biases can guarantee learnability offers valuable insights into prior work. The experiments are also well designed, particularly in leveraging CNN update rules that resemble those of automata. I would like to raise my evaluation accordingly.

---

### Official Review · Reviewer_uiey · 2025-06-30

**Clarity:** 2
**Significance:** 1
**Originality:** 2
**Rating:** 2
**Confidence:** 4

**Summary:**

This paper investigates how structural constraints inspired by physics (i.e., locality, symmetry, and coarse-grained observations) can affect the generalization ability of neural networks, especially for the convolutional neural network (CNN) structure. They apply the CNN models to the cellular automata (CA) for predicting their future state and study how much spatiotemporal locality or localized initial conditions affect the prediction accuracy.

**Questions:**

**Questions & Suggestions**

***Major***
1) I am not convinced that the specific CNN architecture proposed in Section 3.5 is necessary or sufficiently justified for the prediction task. The authors construct a deep CNN with $T$ stacked blocks, one per timestep, but provide no ablation or comparison against simpler alternatives. As I understand it, the key structural priors relevant to this task are spatial locality and translation invariance, both of which are naturally captured by standard CNNs regardless of depth. Therefore, it is unclear whether the depth and timestep-aligned design contribute meaningfully to the learning performance. I believe the authors should provide evidence that their findings are robust across different CNN architectures, or at least clarify whether the proposed architecture is critical to the observed results.

2) In Section 4.1, the authors present a correlation between model accuracy and perturbation sensitivity (PS), showing that accuracy decreases as PS increases, and this effect is amplified with longer time horizons $T$. However, it is unclear what novel insight this provides about the inductive bias of neural networks. The observed trend appears to be a direct and expected consequence of the nature of dynamical systems: greater $T$ naturally allows more time for small perturbations to propagate and amplify, especially in systems exhibiting chaotic behavior. Therefore, while the result is valid, it does not, in itself, reveal anything specific about the network’s inductive bias beyond the trivial fact that sensitive systems are harder to predict over time. I would encourage the authors to clarify what new understanding—if any—about inductive bias is gained from this observation.

3) Following the previous question, a similar concern arises in Sec. 4.2 regarding spatial coarse-graining.  As in the temporal case, this result appears to reflect a trivial reduction in task complexity: coarse-graining removes fine-scale information and effectively simplifies the learning target. It is therefore unclear how this observation supports any specific claim about the inductive bias of neural networks.

4) In Section 4.4, the authors analyze the accuracy of predictions under localized initial conditions and observe that CNNs perform better near the boundary of the evolved structure, with accuracy dropping significantly in the interior. However, this outcome seems to follow directly from the nature of spatial information propagation in cellular automata: boundary regions are less entangled and influenced by fewer interactions, whereas interior regions result from many complex, compounded updates. Thus, the declining accuracy with distance from the edge appears to be a trivial effect of spatial depth in dynamical evolution, rather than a meaningful insight into the neural network’s inductive bias.

***Minor***
1) Page 5: In the description of localized initialization, is the border width $T$ the same as the prediction timestep $T$? Please clarify.
2) Figures 2 and 3: The scatter plots are visually cluttered. Consider using transparency to better reveal overlapping points.
3) Please specify how many steps are used to evolve a configuration before it is used as input. Additionally, how many steps are used to evolve a configuration, and how many steps are required to reach a stationary state?

**Ethical Concerns:**

["NO or VERY MINOR ethics concerns only"]

**Final Justification:**

Despite the authors’ thorough responses to the reviewers’ comments, my major concerns regarding its contribution and non-triviality (Questions 2, 3, and 4) remain unaddressed. I will therefore maintain my original score.

**Limitations:**

Yes.

**Paper Formatting Concerns:**

No.

**Quality:**

2

**Strengths And Weaknesses:**

**Strengths**
1) The use of randomly sampled CAs with physical constraints as testbeds is creative and allows a controlled study of neural network priors
2) The extensive numerical simulations are performed to validate their claims.


**Weaknesses**
1) Only 2D binary CAs are considered, which are abstract and don’t directly reflect real-world systems.
2) The paper is entirely empirical; no theoretical justification is provided for the observed relationships.
3) The goal and the actual contribution of this paper are not clear. I guess that this paper studies how the physical constraints affect the neural network performance, where the network structure intrinsically embodies such constraints as inductive bias. However, the provided evidence is not sufficient to support their claims or seems trivial.

---

> ### Author Rebuttal · Authors · 2025-07-31
>
> We thank the reviewer for their thoughtful feedback and critical observations. Their comments highlighted areas needing clearer exposition, particularly regarding the significance and non-triviality of our contributions.
>
> Our primary aim is to systematically characterize neural networks' inductive bias limitations, specifically examining how complexity impacts learning, the sharpness of these impacts, and the underlying mechanisms. The clarity of this purpose will be improved in our revision.
>
> #### Contribution and Non-Triviality
>
> We understand the reviewer’s perspective that it seems like we only show that complex problems are harder to learn. However, our contribution lies precisely in quantifying and visualizing the sharp thresholds and mechanisms behind these intuitive statements, and how they relate to dynamical systems on short time horizons. In particular our results imply that limited sensitivity to inital conditions (as measured by PS) is not a sufficient condition for learnability on longer time-frames and natural data therefore necessarily exhibits some additional properties which make them learnable.
>
> **Temporal CG:**
>
> The reviewer correctly notes that longer time horizons allow perturbations to amplify. We are aware of this fundamental property of dynamical systems. Indeed, this is precisely why we designed our experiment to measure the Perturbation Sensitivity (PS) not of the single-step rule, but of the entire T-step map, $f^T$. By doing so, we directly account for the compounded effect of the dynamics over the full time horizon.
>
> Even then, we find that learnability collapses at a sharp threshold correlated with the PS of the final map. Further, this learnability cliff becomes even sharper for larger T. This suggests that for a dynamical system to remain learnable over long horizons, its effective T-step function must become progressively simpler (i.e., have a lower PS). The network's inductive bias imposes a much stronger simplicity constraint on functions with long causal chains, a finding that goes beyond the simple intuition that chaotic systems are hard to predict.
>
> **Spatial CG:**
>
> The spatial coarse-graining experiments are not meant to show that simpler tasks are easier. They are designed to test whether the network can leverage simplified, low-frequency information even when it has no knowledge of the underlying data. The finding that relative accuracy gain _decreases_ with more coarse-graining (as the baseline becomes stronger) is an interesting result, showing that the network does not automatically benefit from a simpler signal more than classical machine learning algorithms do.
>
> **Localized Inputs:**
>
> Regarding localized initial conditions, our intent was precisely to illustrate neural networks’ inductive bias toward low-frequency or boundary-dominated signals, showcasing that even in complex functions the network is able to identify simpler sub-functions, prioritizing them for learning. Higher accuracy at boundaries directly demonstrates the network's spectral bias, as boundary regions inherently exhibit simpler, lower-frequency structure than the more complex interior regions.
>
>
> We will clarify these points explicitly in our revised introduction and discussion.
>
> #### Justification for CNN Architecture
>
>
> The T-block CNN design ensures sufficient representational capacity, eliminating representation issues and focusing explicitly on inductive bias and optimization challenges. Reducing the number of convolutional layers would mean that the network cannot use all relevant information, while deeper CNNs would see irrelevant data points. We could however use different kernel sizes, thereby reducing the depth of the CNN. This would result in a different issue: We no longer have a guarantee that the networks are able to represent the specific function due to the often irreducible nature of Automata noted by Reviewer rBzC, and we loosen our prior implicit in the architecture, as we noted in response to Reviewer u7G5. In our Revision, we will take more time to motivate this architectural choice in-depth.
>
> Nevertheless, we also provide additional experiments with ViT and FFN architectures, also discussed in our response to Reviewer u7G5, to further demonstrate the robustness and generality of our findings, which we will also incorporate in our Revision.
>
> #### Minor Clarifications
> - **Localized Initialization (Border Width):** Yes, the border width $T$ is the same as the prediction timestep $T$. This is an intentional choice to ensure that information from the random central patch cannot propagate beyond the absolute edge of the grid. We will clarify this in the revision.
>
> - **Plot Clarity:** Thank you for the suggestion. The figures already contain some transparency, but we will tweak the values to make the plots easier to interpret.
>
> - **"Naturalized" States:** For these experiments, we evolve the initial random configurations for a small, fixed number of steps (specifically, 12 timesteps). The goal is not to reach a true stationary state (which may not exist or be reachable for chaotic rules), but simply to move the input distribution away from uniform noise and towards the system's attractor. We will make this procedure and its motivation more explicit.

---

> ### Comment · Reviewer_uiey · 2025-08-04
>
> I appreciate the authors’ responses, especially their clarifications regarding architecture and their additional experiments, which address my major question 1 and minor suggestions. However, several core concerns remain unresolved.
>
> **Contribution and Non-Triviality**
>
> The authors state that their contribution lies “precisely in quantifying and visualizing the sharp thresholds and mechanisms” that govern learnability. However, it remains unclear what is meant by “sharp thresholds” and “mechanisms.” If this refers to the observed drop in accuracy beyond certain PS values at higher $T$, the paper neither quantifies where such thresholds occur nor explains why they emerge. These appear to be empirical trends, not rigorously defined transitions or explained phenomena.
> Furthermore, the claim that low perturbation sensitivity (PS) is not sufficient for learnability is unsurprising. That a function can be low-sensitivity yet still hard to learn—due to depth, nonlinearity, or optimization difficulties—is already well known. Without identifying what is sufficient, this result does not offer new insight into inductive bias. As it stands, it reiterates expected properties of learning dynamics rather than uncovering new ones.
>
> **Temporal \& Spatial CG**
>
> These points relate to my major questions 2 and 3. The observed performance degradation with longer time horizons and the modest accuracy gains with increased spatial coarse-graining are explained well by task complexity alone. The authors do not convincingly demonstrate that these behaviors reflect anything deeper about inductive biases beyond what is already expected (e.g., long-range dependencies are harder, coarse-grained tasks are easier).
> In particular, the observation that CNNs do not significantly outperform simple baselines under spatial CG may be true, but this is not clearly linked to a limitation of CNN priors or architecture. The result is more a confirmation that simplifying a task makes it easier for all methods—not a new finding about network bias.
>
> **Localized Inputs**
>
> I agree that higher accuracy near boundaries is consistent with known spectral bias in CNNs. However, I remain unconvinced that the results clearly demonstrate spectral bias in this setting. As noted in my original question, boundary dynamics are simpler and less entangled, which could trivially explain the observed accuracy drop toward the interior. The authors do not explore or rule out this alternative explanation, nor do they quantify frequency content to support a spectral interpretation.
>
> Even if the spectral bias explanation holds, it aligns with prior well-established observations. The authors do not provide evidence that this is a new or surprising result, nor do they connect it to broader theoretical or architectural insights.
>
> **Conclusion**
>
> While the empirical work is thorough, the central claims about inductive bias remain underdeveloped.
> I will therefore maintain my original score.

---

### Official Review · Reviewer_YnUM · 2025-07-03

**Clarity:** 3
**Significance:** 2
**Originality:** 3
**Rating:** 5
**Confidence:** 4

**Summary:**

The paper links neural network inductive bias to three structural traits of discrete dynamical systems, locality, symmetry, and input state's entropy, by training a fully convolutional network to predict random outer-totalistic 2-D Cellular-Automaton rules defined on a $3\times3$ Moore neighbourhood with $2^{18}$ possibilities. The authors vary three experimental factors: (i) temporal coarse-graining, (ii) spatial pooling via majority vote, and (iii) the entropy of the initial state (fully random, ``naturalised,'' or localised borders).
Key findings are: (1) convolutional locality and symmetry alone are insufficient for learnability once temporal depth or spatial pooling compound the rule; (2) a rule’s Perturbation Sensitivity (average output change when a single input bit is flipped) is strongly and negatively correlated with predictive accuracy, making it a useful hardness indicator; and (3) low-entropy inputs can mask rule complexity, so input statistics critically shape apparent learnability.

**Questions:**

1. Can you share training-loss curves or variance across random seeds to show whether failures arise from optimization getting stuck or from fundamental representational limits?
2. Would you add accuracy vs iteration (or vs number-of-examples) plots for further clarification?

**Ethical Concerns:**

["NO or VERY MINOR ethics concerns only"]

**Final Justification:**

The authors have included new types of neural network architecture, additional experiments using ViT and FFN, and numerically shown that their claim generalizes over different types of architecture. They also have clarified my questions in training dynamics. Thus, considering their response and the original paper's contribution in theoretical perspective, I increased my score.

**Limitations:**

Some of the limitations are:

1. Only a single CNN variant is tested. A short discussion or comparison experiment with different type of layer could strengthen the paper.

2. Accuracy is reported only after a fixed 4 096 gradient steps. Presenting learning curves or sample-efficiency plots, and noting that conclusions might change with other optimizers or longer training could clarify the scope of the findings.

**Paper Formatting Concerns:**

There are no paper formatting concerns.

**Quality:**

3

**Strengths And Weaknesses:**

Strengths:

1. Carefully constructed “toy universe.”
   Random outer-totalistic 2-D cellular automata on a 3 × 3 Moore neighbourhood
   (2^18 possible rules) give a large yet tractable rule space that isolates the
   effects of locality and symmetry.

2. Motivation of using CNN architecture.
   A fully convolutional network with 3 × 3 and 1 × 1 kernels mirrors the CA’s
   local, translation-invariant update rule and is expressive enough for
   single-step prediction.

3. Comprehensive discussion on experiment factors.
   The study varies three factors: (i) temporal depth (T = 2–7), (ii) spatial
   pooling, and (iii) initial-state entropy, revealing how each
   dimension shapes learnability.

Weaknesses:

1. While the empirical study is convincing, the paper would be significantly stronger with at least a sketch of a capacity bound, sample-complexity bound, which are classical concepts related to learnability.

2. Limited real-world connection.  Experiments are confined to 2-D binary cellular automata; higher-dimensional or continuous-valued systems with physical constraints (such as, conservation
   laws) remain unexplored.

3. Only a CNN is evaluated. Exploring models with different spectral or
   global-context biases, such as Fourier-feature layers (like the work by Tancik et al., 2020 [1]) could reveal whether the observed limitations are CNN-specific.

4. Missing training dynamics.
   The study reports accuracy after the training; the absence of learning
   curves obscures optimisation behaviour, convergence speed, and potential
   over- or under-fitting.

[1] Tancik, Matthew, et al. "Fourier features let networks learn high frequency functions in low dimensional domains." Advances in neural information processing systems 33 (2020): 7537-7547.

---

> ### Author Rebuttal · Authors · 2025-07-31
>
> We sincerely appreciate the reviewer's insightful summary and constructive feedback. We are encouraged by the positive evaluation and clear suggestions. We conducted additional analyses addressing training dynamics, architectural diversity, and will clarify limitations.
>
> #### Training Dynamics: Optimization vs. Representation
>
> We agree that reporting only final accuracy could obscure interesting training dynamics. We decided to not include training dynamics due to limited space, as we are mostly interested in the final accuracy and wanted to keep our focus there.
>
> However, we do have investigated training dynamics, which show different behavior depending on the specific PS region. Low and High PS functions are simple: They either fit the function very quickly or become stuck in a local minimum. Functions close to the "cliff" where performance starts to drop compared to baseline often exhibit more complex behavior: They quickly increase their accuracy in the first few iterations and then become stuck for a variable amount of time in a local minimum before suddenly increasing their accuracy again.
>
> We will include representative training curves comparing accuracy vs iteration in our appendix to illustrate this clearly.
>
> Regarding the question of longer training or different optimizers, we did sweeps with fewer rules to select our specific hyperparameters. Other choices in optimizers or training times did not lead to qualitatively different results in these experiments.
>
> #### Architectural Diversity
>
> We agree with the importance of testing diverse architectures. As noted in our response to Reviewer u7G5, we have conducted additional experiments using ViT and FFN architectures that, broadly speaking, confirm the generality of our results (please see discussion there for details and subtleties; we do see deviations on small T).
>
> We also appreciate the suggestion to try Fourier-space learning; changing the basis for the input is of course a common technique to overcome the low frequency-bias. At this point, we can report that the fully-connected networks, as used in our additional experiments (see answers to u7G5), remove spectral bias in the input domain still lead to the same results for medium to large T (and worse results for small T). Just for clarity: Using a temporal basis transformation in a spectral domain (sampling along the time direction) could probably have a significant impact, but this is at this point outside the scope of our setting (we only consider predictions over fixed time windows).
> #### Strengthening Theoretical Grounding
>
> We appreciate the suggestion to connect our empirical findings to formal complexity measures. While deriving formal bounds is beyond this study's empirical scope, we will explicitly frame PS as a practical empirical measure of function complexity, directly related to the known spectral bias of neural networks [41]. Our revisions will clearly articulate this relationship and emphasize that classical constraints, while necessary, are insufficient to explain generalization performance of neural networks on real data.
>
> #### Limitations and Real-World Connection
>
> We acknowledge the limitation of our simplified "toy universe" approach. This design intentionally isolates fundamental principles like locality and complexity. We agree that higher-dimensional or continuous-valued systems exhibiting conservation laws is an interesting avenue of exploration that sadly did not fit into the scope of this work (we deliberately excluded canonical laws as there is no canonical, easy to justify analogon in the discrete setting).

---

> ### Comment · Reviewer_YnUM · 2025-08-08
>
> I appreciate the authors for their detailed response. Their response have addressed my concerns (especially in terms of architectural diversity and strengthening theoretical grounding), and my questions on training dynamics. Considering both the clarification from the authors and the original work, I will increase my score.

---

### Official Review · Reviewer_u7G5 · 2025-07-03

**Clarity:** 4
**Significance:** 2
**Originality:** 4
**Rating:** 3
**Confidence:** 3

**Summary:**

This paper proposes a dataset based on cellular automata (CAs) as a toy model to probe the structures convoluational neural networks (CNNs) learn.
They sweep over of a large number of rules, vary the prediction horizon, and toggle three knobs—temporal coarse-graining, spatial pooling, and the entropy of initial conditions.
Performance is then correlated with a perturbation-sensitivity (PS) score that measures how fast a single-bit flip spreads through the CA.
Their main result is a sharp learnability cliff: once PS crosses a threshold, accuracy collapses beyond a short horizon, even though locality and symmetry still hold; when the initial state is localized, the network recovers the coarse boundary of emerging patterns but misses interior detail, hinting at a simplicity bias toward low-frequency structure.

**Questions:**

1. How robust is the PS–accuracy cliff to the choice of model?  Have the authors tried a transformer, an RNN, or even a shallow neural operator? Given that the current grid search reportedly took only $\sim 15$ GPU-hours, a quick sweep over a few additional architectures feels tractable and would clarify whether the cliff is CNN-specific.
2. Can the authors sketch a principled argument—e.g.\ via NTK eigenvalues, spectral-bias calculations, or a Lyapunov-style bound—that explains \emph{why} high-PS rules break generalisation beyond a short horizon? Even a toy calculation for a simplified CA would strengthen the empirical story.
3. Besides majority and logistic regression, could the authors add a simple k-nearest-neighbour or kernel predictor? Such baselines would calibrate how much of the performance gap is due to deep learning versus the intrinsic difficulty of high-PS rules.

**Ethical Concerns:**

["NO or VERY MINOR ethics concerns only"]

**Limitations:**

Yes

**Quality:**

4

**Strengths And Weaknesses:**

**Strengths**:
To my knowledge, this is the first systematic link between a concrete complexity metric (PS) and learnability across such a large family of dynamical rules; the empirical sweep is thorough and well documented.
Experiments are extremely well documented: hyperparameters, runtimes, and error bars are clearly laid out, and the dataset is open-sourced.

**Weaknesses**:
All experiments use a single, vanilla CNN; there is no discussion about whether the PS cliff is architecture-dependent.
The study is purely empirical—there is no theoretical account of why PS governs the cliff, nor an attempt to tie the phenomenon to NTK or spectral-bias analyses.
Binary CAs are a long way from real-world signals, so the immediate impact feels limited unless a similar story holds for more realistic dynamics.
No baselines beyond majority/logistic regression are reported; a simple k-NN or kernel predictor could help calibrate how hard the task really is.

---

> ### Author Rebuttal · Authors · 2025-07-31
>
> We sincerely thank the reviewer for their detailed and insightful feedback.
>
> We will now address the three main areas of concern: (1) model architecture dependence, (2) the need for a theoretical explanation for the learnability cliff, and (3) the impact of stronger baselines.
>
> #### 1. Generality of the PS-Accuracy Cliff wrt. different architectures
>
> We agree that this is an important aspect. We would like to explain the rationale behind the CNN design and then address alternative architectures:
>
> First of all, every architecture encodes specific prior knowledge about the functions to be modeled. When injecting such knowledge, our experimental approach requires us to ensure that the problem remains solvable in principle (sufficient training data to identify the function) while not becoming trivial (requiring an additional, preexisting inductive bias to the function). If the first condition were violated, the experiment would become meaningless, as one cannot guess random noise. If the second condition were violated, fitting would become trivial. Our CNN architecture tries to strike this balance by already accounting for the fact that the function can be separated into concatenations of several local functions with small receptive field, but we do not prescribe that they are identical. Different architectures can strengthen or weaken these prior assumptions. A recurrent network (with iterations corresponding to time steps) would additionally assume uniformity of these local functions, which - while also correct - would make the task trivial. The opposite, removing the CNN structure and just applying a network to the full receptive field relevant for all $T$ time steps, however, seems interesting, and we have picked up the suggestion and tried it out: If we decrease prior knowledge by using a fully connected feed forward network (optionally with attention), we lose the prescribed locality. We would expect that this invariably decreases test performance, but we should still see a learning effect. To test this, we conducted new experiments with a simple Vision Transformer (ViT) and a standard Feed Forward Network (FFN).
>
> We obtain interesting additional results: For larger prediction horizons (T≥4), we observe accuracies and gains over baseline that qualitatively match those of the CNN architecture. In particular, all three architectures exhibit a similar performance cliff with increasing PS. For short timescales (T=2,3), the CNN is able to achieve perfect accuracies for low PS, with a drop-off for larger PS depending on T (as shown in the paper). The FFN/ViT architectures do not show the perfect matching regime but rather maintain a drop-off in accuracy with PS with the same shape as for larger T, just with a slightly later onset. FFNs and VITs behave very similarly, with slightly larger training variance for ViTs. This experiment shows that the ability for perfect fitting at small T is indeed induced by the CNN-assumption, but the network can only “utilize” the prior knowledge for small T=2,3 (as one probably would intuitively expect). For medium to long range T, our results are effectively interchangeable between all three architectures (CNNs, FFNs, ViTs).
>
> We would like to thank the reviewer(s) for this suggestion – including this experiment in the revision will certainly make our study more well-rounded; nonetheless, the new insights do not contradict the main findings of the original submission.
>
> #### 2. Argument for the PS Cliff
>
> We appreciate the reviewer’s request for a theoretical explanation. While a full formal proof exceeds this paper's scope, we can offer a principled argument linking our findings to established results.
>
> In our findings, Perturbation Sensitivity (PS) acts as an empirical measurable proxy for complexity. It is closely related to the frequency spectrum of $f^T$. High PS implies that small variations in input lead to rapid fluctuations in output, while low PS implies a more “smooth”, i.e., low-frequency behavior. This connection has already been made more precise in related work, as discussed in Section 2 of our paper (in particular, [41] connects discrete sensitivity to NTK eigenfunctions of transformers).
>
> In essence, PS thus depicts the established spectral bias of neural networks, a preference for learning low-frequency functions. Our observed PS cliff reflects this bias empirically: once PS crosses a critical threshold, the function fluctuates too strongly, overwhelming for the model and causing generalization failures.
>
> While this is a well known result, our work highlights additional nuances: The accuracy drop occurs earlier for higher timesteps, even when we measure PS for the full T timestep function. Thus, for dependencies on larger timescales, some notion of complexity unrelated to sensitivity increases. This leads to the probably most interesting and important finding: While sensitivity is strongly correlated to learnability, it still fails to capture the prior embedded in deep networks fully – there is still something important left to discover. More generally, our paper establishes experimentally that the established principles and measures do not yet capture the true nature of the inductive bias (within the constraints of the discrete model system, of course).
>
> We will clearly articulate this connection in our revised discussion.
>
> #### 3. Additional Baselines
>
> In our work we have mostly used majority vote baselines, as they provide a simple interpretation of uniform prediction and thereby showcase a failure of the network in learning anything except the most basic of patterns. However, as this has been a shared concern between several reviewers, we realize that we should have discussed this more clearly.
>
> We have compared the results using different baselines (majority vote, logistic regression and after this review, added kNN classifiers). Both kNN and logistic regression classifiers rarely outperform majority vote baselines in our experiments. Importantly, when plotting the results in terms of performance over base-line (Fig 2/3 bottom), the results look qualitatively the same for all three choices. We believe that this is due to the prediction tasks being very hard to solve in pixel space.
>
> In our Revision we will make this more clear, and add comparisons using both logistic regression as well as kNN classifiers in the appendix.

---

> ### Comment · Reviewer_u7G5 · 2025-08-06
>
> Dear Authors,
>
> I thank you for your detailed response, in particular your detailed follow-up experiments with other architectures.
>
> I maintain reservations about the impact of this work due to its very simplistic setting. As other reviewers have pointed out, while this work introduces a nice toy problem, it does not shed significant light on the nature of inductive bias in neural networks. The authors introduce PS as a proxy for complexity, which anti-correlates with learnability (and is intriguing in part because it shows a cliff/phase transition with regard to learnability). However, all of these results are very specific to this simple setting and strictly empirical—it is not clear how this notion of complexity can generalize meaningfully to any other setting. Moreover, no theoretical basis is provided for any of the observed results, which could at least invite generalization in a follow up work. (The authors do connect the relationship between PS and learnability to spectral biases of NNs in their response above, but it is not clear to me how this motivates any generalizable insights unique to PS, or the authors' investigation more broadly.)
>
> I laud the authors for their detailed empirical investigation. However, without broadening the impact of this work, I will maintain my score.

---

### Decision · Program_Chairs · 2025-09-17

**Decision:**

Reject

**Comment:**

This submission asks whether the usual priors we build into neural networks (locality, translation symmetry, and coarse-grained observation) make dynamics learnable? The authors create toy universes from two dimensional cellular automata, train a basic convolutional model while varying the prediction window, spatial pooling, and the entropy of the starting state, and report a sharp accuracy drop as a rule’s perturbation sensitivity rises; with localized starts the network recovers coarse boundaries but not interior detail.

The empirical study is careful and well documented, yet reviewers noted key limitations remain and outweigh the merits. The central claims about inductive bias are purely empirical and largely restate expected phenomena without a principled account of why any apparent threshold emerges or where it lies. No capacity or sample complexity bounds or mechanistic links through tools such as neural tangent kernels or spectral analyses are provided to move perturbation sensitivity beyond a descriptive correlate or to identify conditions under which low sensitivity rules remain learnable at longer horizons. The scope is narrow and highly stylized, limited to binary two dimensional automata, with only qualitative links to prior results on computational irreducibility and to more realistic dynamical systems. Although the added architecture sweeps and baselines are welcome, they do not yield actionable insights or demonstrate architectural specificity. After discussion, reviewer scores remain mixed, with two voting accept and two reject. Given the competitive nature of the venue and mixed reviews, I recommend rejection.